# Validation of Two Screening Tools for Detecting Delirium in Older Patients in the Post-Anaesthetic Care Unit: A Diagnostic Test Accuracy Study

**DOI:** 10.3390/ijerph192316020

**Published:** 2022-11-30

**Authors:** Rami K. Aldwikat, Elizabeth Manias, Alex. Holmes, Emily Tomlinson, Patricia Nicholson

**Affiliations:** 1School of Nursing and Midwifery, Deakin University, Geelong, VIC 3220, Australia; 2Centre for Quality and Patient Safety Research, Faculty of Health, Deakin University, Geelong, VIC 3220, Australia; 3Operating Theatre, The Royal Melbourne Hospital, Parkville, VIC 3050, Australia; 4School of Nursing and Midwifery, Faculty of Medicine, Nursing and Health Sciences, Monash University, Clayton, VIC 3800, Australia; 5Department of Medicine, The Royal Melbourne Hospital, The University of Melbourne, Parkville, VIC 3010, Australia; 6Department of Psychiatry, The University of Melbourne, Parkville, VIC 3010, Australia; 7Department of Mental Health, The Royal Melbourne Hospital, Parkville, VIC 3050, Australia; 8Institute for Health Transformation, Deakin University, Geelong, VIC 3220, Australia

**Keywords:** delirium, diagnostic accuracy, screening tools, older people, post-anaesthetic care unit

## Abstract

(1) Background: Delirium is a common complication among surgical patients after major surgery, but it is often underdiagnosed in the post-anaesthetic care unit (PACU). Valid and reliable tools are required for improving diagnoses of delirium. The objective of this study was to evaluate the diagnostic test accuracy of the Three-Minute Diagnostic Interview for Confusion Assessment Method (3D-CAM) and the 4A’s Test (4AT) as screening tools for detection of delirium in older people in the PACU. (2) Methods: A prospective diagnostic test accuracy study was conducted in the PACU and surgical wards of a university-affiliated tertiary care hospital in Victoria, Australia. A consecutive prospective cohort of elective and emergency patients (aged 65 years or older) admitted to the PACU were recruited between July 2021 and December 2021 following a surgical procedure performed under general anaesthesia and expected to stay in the hospital for at least 24 h following surgery. The outcome measures were sensitivity, specificity positive predictive value and negative predictive value for 3D-CAM and 4AT. (3) Results: A total of 271 patients were recruited: 16.2% (44/271) had definite delirium. For a diagnosis of definite delirium, the 3D-CAM (area under curve (AUC) = 0.96) had a sensitivity of 100% (95% CI 92.0 to 100.0) in the PACU and during the first 5 days post-operatively. Specificity ranged from 93% (95% CI 87.8 to 95.2) to 91% (95% CI 85.9 to 95.2) in the PACU and during the first 5 days post-operatively. The 4AT (AUC = 0.92) had a sensitivity of 93% (95% CI 81.7 to 98.6) in the PACU and during the first 5 days post-operatively, and specificity ranged from 89% (95% CI 84.6 to 93.1) to 87% (95%CI 80.9 to 91.8) in the PACU and during the first 5 days post-operatively. (4) Conclusions: The 3D-CAM and the 4AT are sensitive and specific screening tools that can be used to detect delirium in older people in the PACU. Screening with either tool could have an important clinical impact by improving the accuracy of delirium detection in the PACU and hence preventing adverse outcomes associated with delirium.

## 1. Introduction

Delirium is a common and serious neurocognitive disorder that is characterised by fluctuating symptoms related to attention, cognition, and awareness. It can affect up to 50% of older patients during hospitalization [1,2]. Delirium is associated with adverse outcomes, including longer hospital stay, increased risk of subsequent long-term cognitive decline, increased risk of developing dementia, and higher mortality rates [3,4,5]. Further, delirium is associated with increased hospital expenditure [6]. Delirium incidence among adult surgical patients in the post-anaesthetic care unit (PACU) ranges from 4.1% to 45% [7,8]. The wide-ranging rates of incidence highlight the disparity in recognising delirium and indicate that cases may go undetected. Despite these adverse outcomes, evidence suggests that delirium is still under-diagnosed and often misdiagnosed, especially in older hospitalised patients [9,10,11].

The failure to recognise delirium is partially due to a lack of validated screening tools that are sensitive and specific for delirium detection in different clinical settings, and an over-reliance on the subjective judgement of clinicians [12]. Published international guidelines, such as those in Australia developed by the Australian Commission on Safety and Quality in Health Care [13], or in the United Kingdom developed by the National Institute for Health and Care Excellence [14], recommend that people 65 years and older who are admitted to hospital should be screened for delirium. However, the guidelines lacked recommendations for delirium screening tools that are sensitive and specific for older patients in different clinical settings.

The gold standard for the diagnosis of delirium is the Diagnostic and Statistical Manual of Mental Disorders, fifth edition, (DSM-5). Nevertheless, performing assessments with the DSM-5 requires extensive training and is time-consuming. Therefore, validated screening tools are more practical for clinical settings, such as the PACU, as they require little or no training, albeit with the caveat of reduced sensitivity and specificity relative to the gold standard.

There is a limited number of studies examining validated screening tools for delirium detection in the PACU. A recent systematic review examining diagnostic accuracy of screening tools found two tools (3D-CAM and 4AT) demonstrated high sensitivity and specificity values as potential suitable tools to detect delirium in this clinical setting [15]. However, those two tools were only examined among patients who underwent elective surgical procedures, excluding patients with cognitive impairment, dementia, and neurological disorders, making the sample less-representative of general surgical populations [7,16].

Therefore, this study aimed to evaluate the diagnostic test accuracy of the Three-Minute Diagnostic Interview for Confusion Assessment Method (3D-CAM) and the 4A’s Test (4AT) as screening tools for delirium detection in older people in the PACU.

## 2. Materials and Methods

### 2.1. Study Design

This study was designed as a prospective diagnostic test accuracy study. The study adheres to the Standards for Reporting of Diagnostic Accuracy Studies guidelines (STARD 2015) [17,18] (checklist available in Appendix A).

### 2.2. Setting and Participants

The study was conducted at a tertiary care hospital in Victoria, Australia, between 3 July 2021 and 20 December 2021. The hospital has 490 inpatient beds and 140 sub-acute inpatient beds and delivers a comprehensive range of healthcare services across the greater metropolitan area [19]. A recent study conducted at this health care organisation found that the incidence of delirium among inpatients aged 65 years or older was 22.7% [20]. This high incidence emphasises the importance of early screening and identification of delirium. Patients were included in the study if they were aged 65 years or older; scheduled for elective or emergency surgery; admitted to the PACU following a major surgical procedure performed under general anaesthesia; and expected to stay in hospital for at least 24 h following surgery. Patients were excluded from the study if they were incapable of providing informed consent with unavailability of next of kin; transferred directly from the emergency department to the operating theatres for an urgent surgical procedure; unable to communicate in English; had planned admission to the Intensive Care Unit (ICU) following surgery; and scored less than −4, i.e., ‘unresponsive’, on the Richmond Agitation-Sedation Scale (RASS).

### 2.3. Sampling and Recruitment

Following ethical approval (HREC/74575/MH-2021), eligible participants were invited to participate in the study. Informed consent was sought from participants who had the capacity. The next of kin was approached and consulted, in accordance with the provisions of the Mental Capacity Act, with regard to participation in research for those who lacked capacity. Participants were recruited when informed consent was obtained from the next of kin.

Participants were recruited consecutively in the Preadmission Clinic or the Operating Suite Holding Bay between July and December 2021. Participants were assessed for the risk of delirium using a validated tool, the Delirium Risk Assessment Score (DRAS), by an expert researcher in delirium assessment. The DRAS tool cut-off score for the best prediction of patients at high risk is ≥5. The researchers classified participants into two groups: (1) a group with a high risk of delirium, with a score of ≥5, and (2) a group with a low risk of delirium, with a score of <5 [21]. Having high- and low-risk groups was important for testing the diagnostic accuracy of the screening tools [22].

All participants underwent an assessment of cognition preoperatively using the MiniCog tool, to establish participants’ cognition level. The assessments were performed by a single researcher. Information was also obtained from patients’ medical records following cognitive assessment. This comprised demographic and clinical details, including age, gender, admission type, diagnosis of dementia, cognitive impairment, and history of delirium.

### 2.4. Diagnostic Accuracy Procedure

Delirium assessment of the index tests, which refers to the test of interest or the test performance that is being evaluated [23], and the reference standard, which refers to the gold standard, a test or tool that is widely recognised as the best available method to correctly identify the diagnosis of interest [23], were conducted in the PACU at 30 and 60 min following extubation of the endotracheal tube and at the point of discharge from the PACU. The time frame of screening in the PACU is supported in the literature [7,16,24]. Within 30 min of extubation of the endotracheal tube, it was likely that the patient reached an appropriate level of consciousness and was able to respond to commands [16]. Further, the assessment was conducted only if the RASS score was at least −2 (corresponding to light sedation—the patient briefly awakens to voices, eye-opening, and eye contact <10 s). If the RASS score was less than −2, the assessment was postponed to a later time, until it was at least −2. Delirium assessment was also conducted twice daily in the patients’ rooms at approximately 08:00 and 18:00 for the first five post-operative days. Data collection in the postoperative period was based on a recent study by Aldwikat [25], involving 260 patients aged 65 years or older, where it was identified that 100% of post-operative delirium cases developed during the first five post-operative days. Assessment with the reference standard took place within a few minutes of conducting the index tests.

#### 2.4.1. Index Tests

In this study, we employed the index tests of the following delirium screening tools:

##### The 4 A’s Test

The 4 A’s Test (4AT) (see www.the4AT.com) contains four items. The first item assesses alertness. Items 2 to 3 test cognition using the Abbreviated Mental Test-4 (AMT4), which requires the patient to state their age, date of birth, present location, and current year; and attention testing, in which the patient is asked to state the months of the year in backward order, starting with December. Item 4 concerns acute changes in mental status, a core diagnostic feature of delirium. For this item, information may be obtained from different sources, including next of kin, nurses, and health professionals caring for the patient, and from patient medical records. The 4AT scores range from 0 to 12. A score of 0 suggests that delirium or cognitive impairment is unlikely. Scores between 1 and 3 suggest possible moderate to severe cognitive impairment, and a score of 4 or above suggests possible delirium [26].

##### The Three-Minute Diagnostic Interview for Confusion Assessment Method

The 3D-CAM is a derivative and simplified assessment method from the Confusion Assessment Method [27]. The 3D-CAM comprises four features: (1) acute change and fluctuating course, (2) inattention, (3) disorganised thinking, and (4) altered level of consciousness. Each feature is rated as positive or negative for delirium. For delirium diagnosis, the 3D-CAM scoring process requires that features 1 and 2 are both positive; if they are positive, then features 3 and 4 are assessed, and if one of features 3 and 4 is positive, then the 3D-CAM is positive. For completion of the 3D-CAM, information obtained from the patient interview and cognitive testing, and seeking information from medical records, the ward nurse and the next of kin may be required. Details about the 3D-CAM test are provided in the online Appendix A).

##### Reference Standard

The reference standard of delirium was clinical diagnosis carried out using an operationalised format of DSM-5 criteria for delirium diagnosis [2]. The DSM-5 criteria incorporate five features: (1) there is a disturbance in attention and awareness; (2) the disturbance develops over a short period, ‘usually hours to a few days’, represents changes from baseline attention, and tends to fluctuate in severity during the course of the day; (3) there is an additional disturbance in cognition; (4) the disturbances in criteria A and C are not better explained by another pre-existing neurocognitive disorder such as dementia and do not occur in the context of severely reduced levels of arousal; and (5) there is evidence from the history, physical examination or laboratory findings that the disturbance is a direct physiological consequence of another medical condition [2]. DSM-5 criteria details are provided in the online Appendix A).

Assessment of the reference standard was performed by the one researcher who conducted the index tests, an experienced clinician with extensive experience in delirium assessment. The researcher received formal one-on-one training from a board-certified consultant psychiatrist on diagnosing delirium using the DSM-5 criteria. The researcher also performed 20 assessments under the direct supervision of the board-certified consultant psychiatrist prior to commencing the study to guide decision-making with regard to delirium diagnosis to ensure diagnostic accuracy.

To determine the reliability of the researcher, 30 assessments were blindly co-rated by the consultant psychiatrist. Patients were classified into definite delirium (meeting all DSM-5 criteria), possible delirium (‘do not know’ researcher response for one or more of the DSM-5 criteria, with all other criteria met), and no delirium (failing to meet one or more of the DSM-5 criteria).

### 2.5. Randomisation

The order of the two index tests conducted for this study, the 3D-CAM and the 4AT, was randomly allocated according to the computer-generated randomisation method, SAS statistical software, version 9.3 [28]. The randomisation system was web-based and required a personal login and password. Stratified randomisation, with block allocations of 10, was used. Further, randomisation was stratified into high- and low-risk groups. This stratification was to minimise any variation in testing the screening tool success rate due to the differences in risk of delirium between high- and low-risk patients. The DSM-5 was administered to all patients as a reference standard.

Participants were randomised at a 1:1 ratio and were assessed using the 4AT and the 3D-CAM screening tools. Similarly, with regard to the inter-rater agreement, participants were randomised at a 1:1 ratio and were assessed using the reference standard test. Once randomisation had been performed, neither the researcher nor the participant was blinded to the allocation because both were aware of the assessments to be conducted and the order in which they were performed.

### 2.6. Sample Size Calculation

The primary aim of the study was to evaluate the diagnostic accuracy of the 4AT and the 3D-CAM against a reference standard diagnosis, and the sample size calculation was based on this. The power calculation was based on the method recommended by Buderer [29]. Based on previous work on the 4AT and 3D-CAM in the PACU, the sensitivity ranged from 96% to 100%. A minimum sensitivity of 85% was set as the lowest level of acceptable sensitivity for the study, allowing for 95% confidence intervals with no more than 10% width of a two-sided 95% CI. To account for dropouts, a number of enrolled patients higher than 270 was targeted.

### 2.7. Statistical Analysis

Patients’ characteristics and clinical variables are expressed as percentages, frequencies, mean, and median. Continuous data were analysed using the *t*-test, and non-continuous data were analysed using the chi-square test.

Test accuracy statistics were described, including the sensitivity, specificity, positive predictive values and negative predictive values and corresponding to 95% confidence intervals, comparing the two index tests (4AT and 3D-CAM) against the reference standard (DSM-5) criteria for delirium diagnosis. For the 4AT, a predetermined cut-off of ≤4/12 was used, as suggested by the tool developer. The 3D-CAM was dichotomised as positive or negative delirium. A Receiver Operating Characteristic (ROC) analysis was carried out separately on the 4AT and the 3D-CAM, comparing them to DSM-5 criteria for delirium diagnosis. As an index of reliability, the inter-rater reliability between the researcher and the psychiatrist consultant for DSM-5, with a corresponding 95% confidence interval, was calculated using Cohen’s Kappa coefficient (k). A *p*-value of <0.05 was considered statistically significant. Statistical analysis was performed using Stata 13.0 (Stata Corp., College Station, TX, USA).

## 3. Results

### 3.1. Demographic and Clinical Characteristics

A total of 318 patients were available during the recruitment process; 26 patients did not fulfil the inclusion criteria, so there were 292 patients eligible to participate in the study. Of these, 10 (3.4%) declined to participate, 5 (1.7%) and were transferred to the ICU immediately after surgery, 4 (1.4%) had regional anaesthesia, and 2 (0.7%) had their surgery cancelled. This resulted in 271 patients being recruited and screened for delirium over a 6-month period. Figure 1 displays the study flow chart.

Of the participants, 139 were female (51.2%) and the mean age of the study population was 76.9 years (SD = 7.9). Eighty-seven participants (32%) presented with cognitive impairment on admission to hospital (MiniCog score ≤ 2). Ten patients (3.7%) were reported as having dementia on admission. Seventy-three (27%) patients presented with a decline in the ability to perform activities of daily living independently (Katz score < 6). The mean number of pre-existing comorbidities was 5.2 (SD = 2.1). A total of 104 (38%) patients underwent trauma/emergency surgery, and 167 (62%) patients underwent elective surgery. The median anaesthesia time was 138 min (interquartile range (IQR): 88–196 min). A summary of participants’ characteristics with respect to the presence of delirium is presented in Table 1.

The age of patients in the delirium group was significantly older than those in the non-delirium group (M = 83.8, SD = 8.5 vs. M = 75.5, SD = 7.9; *p* < 0.0005). Using Fisher’s exact test, patients with a MiniCog of <3 were more likely to have an episode of delirium (*p* < 0.0001), as were those with dementia. There was no statistical difference with regard to gender and duration of anaesthesia.

### 3.2. Comparison of Test Performance

Using DSM-5 criteria for delirium diagnosis, 16.2% (44/271) of patients had definite delirium, 7% (19/271) had possible delirium, and 76% (208/271) had no delirium.

The 4AT index test assessment was performed with a mean time of 2.8 min (SD = 0.90), whereas the 3D-CAM index test assessment was performed with a mean time of 3.6 min (SD = 0.72).

A ROC analysis was carried out to calculate the sensitivity and specificity values of the 4AT and the 3D-CAM when comparing screening accuracy to the DSM-5 definite delirium, with a cut-off score ≥4 for the 4AT and 3D-CAM positive diagnosis of delirium. The receiver operation characteristic curves of the 4AT and the 3D-CAM are displayed in Figure 2.

The 3D-CAM (AUC = 0.96) had excellent diagnostic accuracy, with a sensitivity of 100% (95% CI 92.0 to 100.0) in the PACU and during the first 5 days post-operatively. Specificity ranged from 93% (95% CI 87.8 to 95.2) to 91% (95% CI 85.9 to 95.2) in the PACU and during the first 5 days post-operatively. For the 4AT (AUC = 0.92), sensitivity was 93% (95% CI 81.7 to 98.6) in the PACU and during the first 5 days post-operatively, and specificity ranged from 89% (95% CI 84.6 to 93.1) to 87.0% (95% CI 80.9 to 91.8) in the PACU and during the first 5 days post-operatively (Table 2).

### 3.3. Inter-Rater Reliability

For inter-rater reliability using the DSM-5 criteria, 30 patients were evaluated simultaneously and independently by the researcher and consultant psychiatrist. The assessments were concordant in 28 out of the 30 assessments using Kappa. The strength of the agreement between the assessors (coefficient) was 0.876 (*p* < 0.001). Interpreting Kappa statistics of inter-rater agreement suggested that a coefficient of 0.81–1.00 is considered almost perfect [30].

### 3.4. Delirium Subtypes

Of the 44 patients found to be delirious, according to the DSM-5 criteria, 6 (13%) presented in the hypoactive state of delirium, 33 (75%) in a mixed state and 5 (11%) in the hyperactive state.

## 4. Discussion

In this study, 16.2% of the patients developed definite delirium using DSM-5 criteria in the PACU. The evaluated screening tools demonstrated high sensitivity and specificity in a broad surgical population of older patients undergoing elective or emergency surgery. Both the 3D-CAM and the 4AT were found to be suitable tools to detect delirium in the PACU, and both demonstrated high sensitivity and specificity in this study.

The incidence of postoperative delirium in older surgical patients in the PACU was 16.2%, which is lower than 45% as reported in Neufeld et al.’s study [8]. A possible explanation for this low incidence in comparison to Neufeld et al.’s study is that we assessed patients in the PACU at 30 and 60 min following extubation of the endotracheal tube, and at the point of discharge from the PACU, so we were able to identify any possible changes in cognition and the presence of delirium features. Neufeld et al. only conducted one assessment in the PACU once the patient reached a score of ≥9 on the Aldrete scoring system. Further, patients who were transferred to the ICU were included in their sample. This ICU population is known to have a high incidence of delirium, whereas in our study, we excluded patients who were transferred to the ICU. Nevertheless, the incidence of postoperative delirium in our study is consistent with the ranges described in the literature for postoperative delirium in patients admitted to the PACU.

Neither the 3D-CAM nor the 4AT have previously been validated in the PACU in people with pre-existing cognitive impairment and dementia, which are common in the general hospital setting. A recent study using 3D-CAM, which excluded those with pre-existing cognitive impairment, reported a sensitivity of 100% and specificity of 88% [16]. These figures are similar to those found in the current study, suggesting that the application of the 3D-CAM in clinically representative populations does not reduce its accuracy. This is of practical significance since reductions in sensitivity represent cases of delirium being missed, and reductions in specificity lead to false positives or cases of delirium being incorrectly identified and the potential for unnecessary investigations or inappropriate triage to delirium pathways.

One recent study by Saller et al. [7] reported a sensitivity of 95.5% and specificity of 99%. It is important to note that while their sensitivity and specificity values are higher than the diagnostic accuracy values in our study, the 4AT tool in Saller et al. study was conducted among people of 18 years and above and excluded patients with cognitive impairment and dementia; hence, delirium incidence in their study was 4.1%. In contrast, our study included an older population at risk of delirium, with a mean age of 76.9 years, and those with cognitive impairment and dementia, who demonstrated a higher definite delirium incidence of 16.2%. Our study also included 50% of the population at high risk of delirium (DRAS score ≥ 5), which may have caused the screening tool to have slightly lower specificity values. Our finding regarding the diagnostic test accuracy of the 4AT is also supported by two previous studies. One study examined the 4AT in older patients admitted to the geriatric medical unit reported a sensitivity of 86.7% and reasonable specificity of 70% [1], while another study examined the diagnostic test accuracy of the 4AT in older patients admitted to an acute geriatric ward and rehabilitation department, and reported a sensitivity of 89.7% and specificity of 84%, which are similar to our reported values [26]. Furthermore, a recent study by De et al. emphasised the high values of diagnostic test accuracy of the 4AT when conducted in older patients admitted to geriatric and orthogeriatric wards, where a sensitivity of 87% and specificity of 80% were reported [31].

Although the 3D-CAM and the 4AT tests have similar diagnostic accuracy results, both have advantages and disadvantages. The 4AT is simpler and quicker to complete and can be administered without training. However, it has a higher number of false-positive results (not diseased but tested positive) compared with the 3D-CAM. On the other hand, the 3D-CAM, developed by Palihnich et al. [32], requires minimal training, with a training manual available (www.hospitalelderlifeprogram.org) to assist clinicians when assessing a patient for delirium.

Some limitations of the study need to be acknowledged. It was conducted at a single site in a tertiary care metropolitan hospital, which may limit the generalisability of the findings. However, we believe the findings are applicable to a broader patient population because of a wide range of selection criteria. Moreover, we did not assess the severity of delirium, as the focus of the study was to evaluate the diagnostic accuracy of the tools in all patients with and without delirium. Further, it is essential that screening tools for delirium detection are also suitable for use in patients with dementia; however, in our sample, only 10 patients had a known diagnosis of dementia on admission to hospital, which could be underrepresented in this cohort. Although these patients with dementia were diagnosed as having delirium, it is difficult to establish how well the 3D-CAM and the 4AT can distinguish the presence of delirium. Moreover, most patients who were diagnosed with delirium had some degree of cognitive impairment at baseline; thus, it is difficult to establish how well the screening tools distinguish delirium presence in patients with cognitive impairment. The incidence of delirium in our study was considerably low (16.2%). A possible explanation is that 62% of patients underwent elective surgical procedures; therefore, this needs to be considered in future studies. Further, the researchers who validated the 3D-CAM and the 4AT are qualified delirium assessors. It is unknown how these tests would perform in detecting delirium if they were conducted by less experienced practitioners.

The index tests and the reference standard were performed by one researcher. To overcome this limitation, we performed stratified randomisation, thus controlling the possible influence of a test that would jeopardise the results of other tests [33]. According to Schulz [34], randomisation produces interpretable and valid results. Additionally, we adhered to the STARD Checklist for Reporting of Diagnostic Accuracy Studies guidelines with regards to conducting index test and reference standard. In addition, the researcher who performed the screening is a qualified delirium assessor and has extensive experience in delirium screening; however, it is unknown how the 3D-CAM and 4AT will perform in detecting delirium when used by less experienced practitioners.

Finally, we did not evaluate feasibility of the screening tools in this study. It is recommended that follow-up studies should be extended to evaluate evidence of the tools comparative performances, feasibility, and nurses’ perceptions in the PACU and surgical wards in order to improve the recognition and identification of delirium.

Despite these limitations, a strength of our study includes enrolling patients with cognitive impairment, known dementia, or other neurocognitive disorders, as well as patients who underwent emergency and elective surgical procedures, which were excluded in other studies [7,8,16,24]. Therefore, our findings are applicable to broader patient populations. In addition, we assessed cognitive impairment pre-operatively to establish the level of cognition and improve the diagnosis accuracy according to the DSM-5 and index tests.

Further, data were collected 30 min after hospital admission to day 5 post-operatively, which may have reduced the diagnostic uncertainty of delirium. In clinical practice, patients are assessed over a period of time, often days, and this cumulative information and observation facilitate the determination of whether delirium is present or not. Most studies validating screening tools for delirium in the PACU were conducted at a single point in time during their PACU stay [7,8,16,24], which may lead to diagnostic uncertainty over time.

Furthermore, inter-rater analysis in this study provided valuable information about the diagnostic accuracy of the study. For inter-rater evaluation, 30 patients were assessed independently by the researcher, who was trained in the DSM-5 delirium criteria and by the consultant psychiatrist at the same time. An agreement coefficient of 0.876 suggests that the researcher can safely apply DSM-5 criteria for delirium diagnosis, and diagnostic accuracy in this study is almost perfect.

Moreover, this study is the first to validate screening tools for delirium detection with a sample consisting of 50% at high risk of delirium and 50% at low risk of delirium. This process further improves the accuracy of the diagnostic tests of the study. According to Ransohoff and Feinstein [35], for diagnostic test accuracy, to rule out a condition, a test should have high accuracy for negative prediction (close to 100%). Consequently, to establish test efficacy for ruling out a condition, sensitivity should also be tested in a broad range of participants with the condition. Similarly, to rule in a condition, a test should have high accuracy for positive prediction (close to 100%). Therefore, a test should be challenged for its specificity in a broad range of participants without the condition [35].

## 5. Conclusions

Both the 3D-CAM and the 4AT are sensitive and specific screening tools that can be used to detect delirium in older people in the PACU. Both tools are simple and quick to administer and therefore can be useful in the prompt diagnosis and treatment of delirium if incorporated into clinical practice. Given the high incidence of delirium in older people in the PACU and the adverse outcomes associated with delirium development, routine screening with either tool will not only improve accurate recognition and detection of delirium, but also have critical implications when caring for older people by improving the quality of care provided and preventing adverse outcomes. Further work on feasibility and nurses’ perceptions of using these tools is required to effectively capture comprehensive data about the utility of these screening tools for delirium in the PACU.

## Figures and Tables

**Figure 1 ijerph-19-16020-f001:**
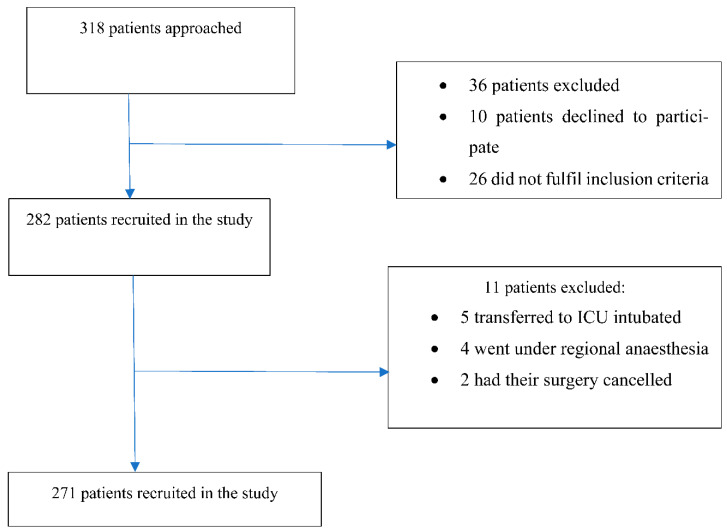
Participants who were included, excluded, and completed the study.

**Figure 2 ijerph-19-16020-f002:**
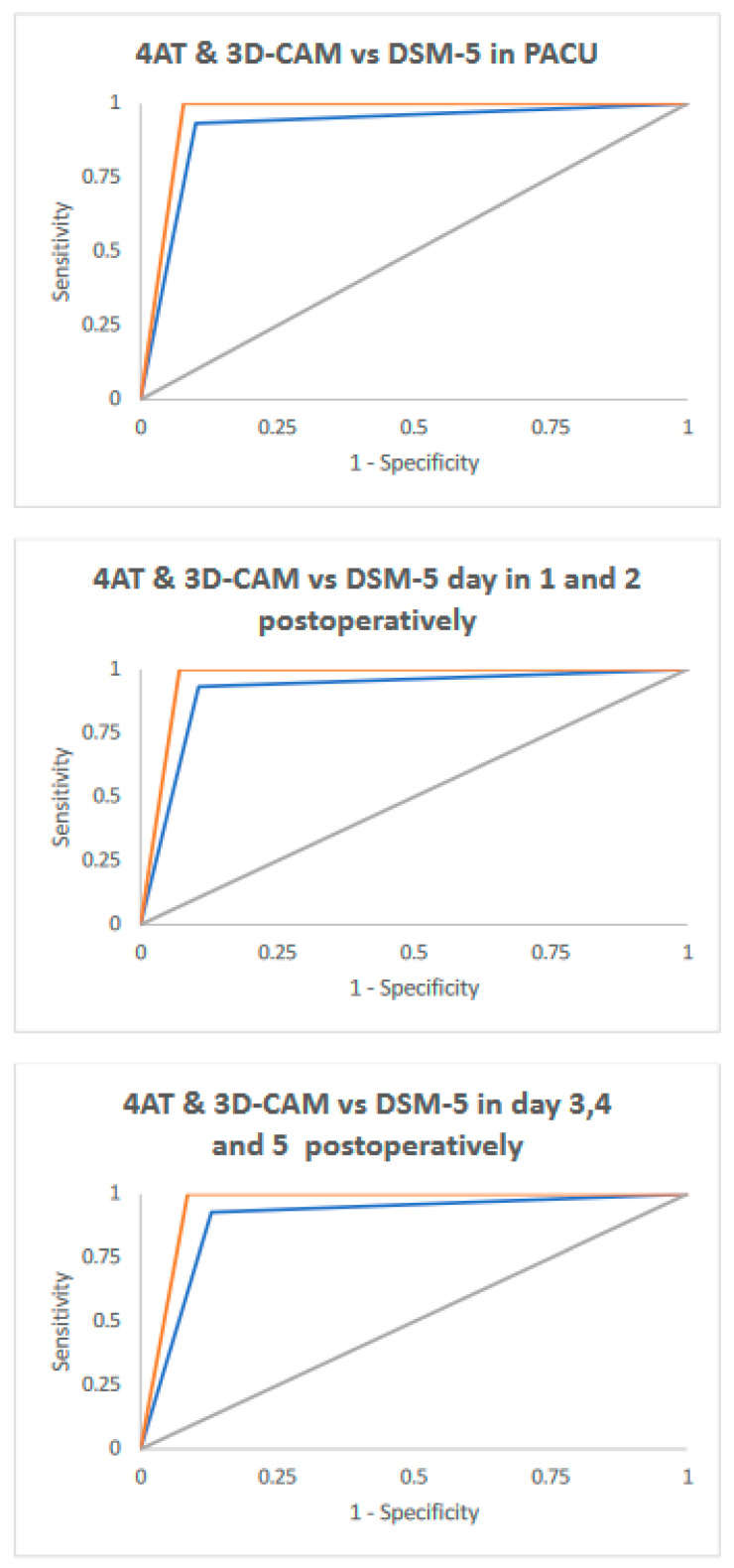
Receiver operation characteristic curve of the 4AT (blue curve) and the 3D-CAM (orange curve) compared with the reference standard (DSM-5).

**Table 1 ijerph-19-16020-t001:** Baseline characteristics of participants by delirium diagnosis DSM-5 (definite delirium).

	All Participants (*n* = 271)	No Delirium (*n* = 227)	Delirium (*n* = 44)	*p* Value
Mean age (years) (SD)	76.9 (7.9)	75.5 (7.9)	83.8 (8.5)	0.0005
Gender				0.423
Male *n* (%)	132 (49%)	113 (50%)	19 (43%)	
Female *n* (%)	139 (51%)	114 (50%)	25 (57%)	
Dementia *n* (%)	10 (3.7%)	0 (0%)	10 (22.7%)	0.001
MiniCog score < 3 (preoperatively) (%)	87 (32%)	45 (20%)	42 (95.5%)	0.001
Mean Katz-ADL score (preoperatively) (SD)	5.3 (0.43)	5.7 (0.53)	4.95 (1.29)	0.001
Mean Charlson comorbidity index score	5.22 (2.1)	5.0 (2.2)	6.4 (2.0)	0.005
Type of surgery				0.085
Orthopaedic (%)	101 (37%)	72 (31.7%)	29 (66%)	
Urology (%)	35 (12.9%)	33 (14.5%)	2 (4.5%)	
ENT (%)	21 (7.7%)	21 (9.2%)	0 (0%)	
Lap-abdomen (%)	20 (7.4%)	19 (8.3%)	1 (2.2%)	
Colorectal (%)	18 (6.6%)	15 (6.6%)	3 (6.8%)	
Plastic (%)	17 (6.3%)	14 (6.1%)	3 (6.8%)	
Thoracic (%)	16 (5.9%)	13 (5.7%)	3 (6.8%)	
Reno-vascular (%)	13 (4.7%)	13 (5.7%)	0 (0%)	
Vascular (%)	13 (4.7%)	12 (5.2%)	1 (2.2%)	
Breast (%)	11 (4.1%)	11 (4.8%)	0 (0%)	
Hepatobiliary (%)	4 (1.4%)	3 (1.3%)	1 (2.2%)	
Cardiothoracic (%)	3 (1.1%)	2 (0.8%)	1 (2.2%)	
Median duration of anaesthesia (min) (IQR)	138 min (88–196 min)	139 min (83–211 min)	136 min (98–167 min)	0.890

DSM-5 The Diagnostic and Statistical Manual of Mental Disorders (DSM-5) criteria for delirium diagnosis; SD: Standard deviation; IQR: Interquartile range. ENT: Ear, nose, and throat; Katz-ADL: Katz Index of Independence in Activities of Daily living.

**Table 2 ijerph-19-16020-t002:** Diagnostic test accuracy values of the 4AT and 3D-CAM in the study cohort of 271 patients. Patients with a definite delirium diagnosis (*n* = 44) were classified as delirium positive compared with all other patients (*n* = 227).

	Tool	Sensitivity (95% CI)	Specificity (95% CI)	PPV (95% CI)	NPV (95% CI)	AUROC (95% CI)
In PACU	4AT	93.2% (81.3–98.6)	89.9% (85.2–93.5)	64.1% (51.1–75.7)	98.6% (95.8–99.7)	0.92 (0.87–0.96)
	3D-CAM	100.0% (92.0–100.0)	93.0% (87.8–95.2)	71.0% (58.1–81.8)	100.0% (98.3–100.0)	0.96 (0.94–0.98)
Day 1 post-operatively	4AT	93.3% (81.7–98.6)	89.4% (84.6–93.1)	63.6% (50.9–75.1)	98.5% (95.8–99.7)	0.91 (0.87–0.96)
	3D-CAM	100.0% (92.1–100.0)	93.0% (88.8–95.9)	73.8% (60.9–84.2)	100.0% (98.3–100.0)	0.96 (0.95–0.98)
Day 2 post-operatively	4AT	93.3% (81.7–98.6)	89.4% (84.6–93.1)	63.6% (50.9–75.1)	98.5% (95.8–99.7)	0.91 (0.87–0.96)
	3D-CAM	100.0% (92.1–100.0)	93.0% (88.8–95.9)	73.8% (60.9–84.2)	100.0% (98.3–100.0)	0.96 (0.95–0.98)
Day 3 post-operatively	4AT	92.9% (80.5–98.5)	87.0% (80.9–91.8)	65.0% (51.6–76.9)	97.9% (94.0–99.6)	0.90 (0.85–0.95)
	3D-CAM	100.0% (91.6–100.0)	91.4% (85.9–95.2)	75.0% (61.6–85.6)	100.0% (97.5–100.0)	0.96 (0.94–0.98)
Day 4 post-operatively	4AT	92.9% (80.5–98.5)	87.0% (80.9–91.8)	65.0% (51.6–76.9)	97.9% (94.0–99.6)	0.90 (0.85–0.95)
	3D-CAM	100.0% (91.6–100.0)	91.4% (85.9–95.2)	75.0% (61.6–85.6)	100.0% (97.5–100.0)	0.96 (0.94–0.98)
Day 5 post-operatively	4AT	92.9% (80.5–98.5)	87.0% (80.9–91.8)	65.0% (51.6–76.9)	97.9% (94.0–99.6)	0.90 (0.85–0.95)
	3D-CAM	100.0% (91.6–100.0)	91.4% (85.9–95.2)	75.0% (61.6–85.6)	100.0% (97.5–100.0)	0.96 (0.94–0.98)

4AT: The 4 A’s Test; 3D-CAM: Three-Minute Diagnostic Interview for the Confusion Assessment Method; PPV: Positive Predictive Value; NPV: Negative Predictive Value; CI: Confidence Interval; AUROC: Area under the receiver operating characteristic curve.

## Data Availability

Data regarding this study can be provided upon request to the corresponding author. Data are not publicly available due to ethical issues.

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
