# Peer review of "Validation of Two Screening Tools for Detecting Delirium in Older Patients in the Post-Anaesthetic Care Unit: A Diagnostic Test Accuracy Study"

_ijerph, 2022, doi:10.3390/ijerph192316020_

Round 1

Reviewer 1 Report

This is generally a methodologically sound paper.  The authors chose 2 well known tests that they validated, using an appropriate gold standard, on a relevant population at moderate risk for delirium (as opposed to a non demented sample).  The appendices were also helpful.  The following are some suggestions for the authors to consider:

In the results section, please provide a description of what surgical procedures patients underwent and some information on co-morbidity & functional dependence.  Since the authors already assessed the DRAS this data should be available and would be of interest to the reader.  It will also help to establish external validity. 

The results are interesting in that although dementia was not an exclusion criteria (which is a good feature of the study), only 10 patients (3.7%) had a known diagnosis of dementia (quite low).  However about 1/3 may have had some cognitive issues at baseline (based on the MiniCog score).  As such, although the intention was to include demented patients, there weren't that many demented patients included.  This should be mentioned as a limitation.  Results also show that everyone who had dementia had delirium.  In this way, the study can't comment on how well these tests distinguish delirium presence in people who have known dementia.  This should also be mentioned as a limitation.  Please correct the typo in Table 1 (% of delirious patients with a low Mini cog score is written as 195%). It would appear that most people who were delirious had some degree of cognitive impairment at baseline (again suggesting that we don't know how well these tests do to pick up delirium in cognitively impaired persons; this should be mentioned as a limitation).

The incidence of new delirium was 16.2%. Although this is higher than in some studies that the authors make reference to, it's still not that high (makes me wonder if there were many emergent or hip fracture surgeries included in the study, another reason to include information on the types and percentages of surgeries included).  The authors do try and include delirium risk assessment in their study which is a good feature of the study, but still overall the delirium incidence was not that high (this may reflect the surgeries included); a comment on this would be helpful.

Finally although the 4AT doesn't require much training (the 3D CAM may require some training), the researchers were well trained in administering these tests. How these tests would perform if they were done by less experienced practioners in not clear from study results.  This should be included as a limitation.

Author Response

28 November 2022

Response to Decision Letter

Dear Professor Paul B Tchounwou

In this document, we have responded to all the reviewers’ comments as addressed in the table below as well as in the revised manuscript. All authors agree with this version of the revised manuscript.

Reviewer comments

Responses to each of the comments

Reviewer 1. ‘‘Comments to the Author’’

This is generally a methodologically sound paper.  The authors chose 2 well known tests that they validated, using an appropriate gold standard, on a relevant population at moderate risk for delirium (as opposed to a non demented sample).  The appendices were also helpful.  The following are some suggestions for the authors to consider:

·      In the results section, please provide a description of what surgical procedures patients underwent and some information on co-morbidity & functional dependence.  Since the authors already assessed the DRAS this data should be available and would be of interest to the reader.  It will also help to establish external validity. 

·      The results are interesting in that although dementia was not an exclusion criteria (which is a good feature of the study), only 10 patients (3.7%) had a known diagnosis of dementia (quite low).  However about 1/3 may have had some cognitive issues at baseline (based on the MiniCog score).  As such, although the intention was to include demented patients, there weren't that many demented patients included.  This should be mentioned as a limitation.  Results also show that everyone who had dementia had delirium.  In this way, the study can't comment on how well these tests distinguish delirium presence in people who have known dementia.  This should also be mentioned as a limitation. 

·      Please correct the typo in Table 1 (% of delirious patients with a low Mini cog score is written as 195%).

·      It would appear that most people who were delirious had some degree of cognitive impairment at baseline (again suggesting that we don't know how well these tests do to pick up delirium in cognitively impaired persons; this should be mentioned as a limitation).

·      The incidence of new delirium was 16.2%. Although this is higher than in some studies that the authors make reference to, it's still not that high (makes me wonder if there were many emergent or hip fracture surgeries included in the study, another reason to include information on the types and percentages of surgeries included). The authors do try and include delirium risk assessment in their study which is a good feature of the study, but still overall the delirium incidence was not that high (this may reflect the surgeries included); a comment on this would be helpful.

·      Finally although the 4AT doesn't require much training (the 3D CAM may require some training), the researchers were well trained in administering these tests. How these tests would perform if they were done by less experienced practioners in not clear from study results.  This should be included as a limitation.

Thank you for the comment.

This info has now been added into the Demographic and Clinical Characteristics section in the

Results section (page 6) and in Table 1, please see below.

Eighty-seven participants (32%) presented with cognitive impairment on admission to hospital (MiniCog score ≤2). Ten patients (3.7%) were reported as having dementia on admission. Seventy-three (27%) patients presented with a decline in the ability to perform activity of daily living independently (Katz score <6). The mean number of pre-existing comorbidities was 5.2 (SD =2.1). One hundred and four (38%) patients underwent trauma/emergency surgery and 167 (62%) patients underwent elective surgery. The full description of surgical procedures is presented in Table 1.

This info has now been added into the limitation section, please see below.

Further, it is essential that screening tools for delirium detection are also suitable for use in patients with dementia, however, in our sample, only 10 patients had a known diagnosis of dementia on admission to hospital, which could be underrepresented in this cohort. Although these patients with dementia were diagnosed as having delirium, it is difficult to establish how well the 3D-CAM and the 4AT can distinguish the presence of delirium.  (page number= 11, paragraph number= 1).

Thank you for the comment.

The typo has been corrected as 42 (95.5%), please see table 1.

Thank you for the comment.

This info has now been added into the limitation section, please see below;

Moreover, most patients who were diagnosed with delirium had some degree of cognitive impairment at baseline, thus, it is difficult to establish how well the screening tools distinguish delirium presence in patients with cognitive impairment. (page number= 11, paragraph number= 1).

Thank you for the comment.

The type and percentage of surgeries has now been added into the result section, please see Demographic and Clinical Characteristic section as well as Table 1, It is also now reflected in the limitation section.

In the limitation section:

The incidence of delirium in our study was considerably low (16.2%). A possible explanation is that 62% of patients underwent elective surgical procedures, therefore, this needs to be considered in future studies.

(page number= 11, paragraph number= 1).

Thank you for the comment.

This info has now been added into the limitation section, please see below;

Further, the researchers who validated the 3D-CAM and the 4AT are qualified delirium assessors. It is unknown how these tests would perform in detecting delirium if they were conducted by less experienced practitioners.

(page number= 11, paragraph number= 1).

Reviewer 2 Report

This is a single-center diagnostic study regarding the accuracy of two delirium screening tests in a cohort of elderly patients in a postoperative care setting. The sample size and methods are adequate. The paper is well written. The results are in agreement with the available literature, and offer data on diagnostic accuracy of two short delirium screening tests among the at-risk subgroup of elderly patients as a novel finding.

I recommend three minor revisions: 

1.) Please rewrite chapter 2.5 on randomization in a clearer style. All patients received all three test, so what exactly does 1:1 randomisation refer to? 

2.) Please carefully perform another spell check, e.g. P6 (10 patients are not 0.04%), table 1 (42 patients are not 195%), table 2 (AUC is not reported as %), P9 (specificity is likely not 0.88%). 

3.) Please discuss all relevant references (e.g. 1, 8, 26) in the context of the present study and perform another current review of relevant literature (e. g. DOI: 10.1002/gps.4615).

Author Response

28 November 2022

Response to Decision Letter

Dear Professor Paul B Tchounwou

In this document, we have responded to all the reviewers’ comments as addressed in the table below as well as in the revised manuscript. All authors agree with this version of the revised manuscript.

Reviewer comments

Responses to each of the comments

Reviewer 2. ‘‘Comments to the Author’’

This is a single-center diagnostic study regarding the accuracy of two delirium screening tests in a cohort of elderly patients in a postoperative care setting. The sample size and methods are adequate. The paper is well written. The results are in agreement with the available literature, and offer data on diagnostic accuracy of two short delirium screening tests among the at-risk subgroup of elderly patients as a novel finding.

I recommend three minor revisions: 

1)    Please rewrite chapter 2.5 on randomization in a clearer style. All patients received all three test, so what exactly does 1:1 randomisation refer to? 

2)    Please carefully perform another spell check, e.g. P6 (10 patients are not 0.04%), table 1 (42 patients are not 195%), table 2 (AUC is not reported as %), P9 (specificity is likely not 0.88%). 

3)    Please discuss all relevant references (e.g. 1, 8, 26) in the context of the present study and perform another current review of relevant literature (e. g. DOI: 10.1002/gps.4615).

1)    Thank you for the comment.

Section 2.5 Randomisation has been reviewed and written in clearer style as below.

The order of the two index tests conducted for this study, the 3D-CAM and the 4AT, was randomly allocated according to the computer-generated randomisation method,  SAS statistical software, version 9.3.28 The randomisation system was web-based and required a personal login and password. Stratified randomisation, with block allocations of 10, was used. Further, randomisation was stratified into high and low risk groups. This stratification was to minimise any variation in testing the screening tool success rate due to the differences in risk of delirium between high and low risk patients. The DSM-5 was administered to all patients as a reference standard.

Participants were randomised in a 1:1 ratio, where they were assessed using the 4AT and the 3D-CAM screening tools. Similarly, with regard to the inter-rater agreement, participants were randomised in a 1:1 ratio, where they were assessed using the reference standard test. Once randomisation had been performed, neither the researcher nor the participant was blinded to the allocation because both were aware of the assessments to be conducted and the order in which they were performed.

2)    Thank you for the comment.

Spell check has been performed and correction made as bellow and in Tables 1& 2:

·      In page 6: Ten patients (3.7%) were reported as having dementia on admission.

·      Table 1 (42 patients are 95.5%) see Table 1.

·      Table 2 (for AUC, % was removed from reporting), for example in PACU AUC now reported as 0.92 instead of 0.92% as was previously reported, see Table 2.

·      P9 (specificity is now corrected to 88% instead of 0.88%).  See Discussion section, second paragraph.

3)    Thank you for the comment.

all relevant references have now been discussed in the context of the present study, also we performed another current review of relevant literature, and also discussed the study (e. g. DOI: 10.1002/gps.4615) in the in the context of the present study as bellow:

The incidence of postoperative delirium in older surgical patients in the PACU was 16.2%, which is lower than 45% as reported in Neufeld at al. study.8 A possible explanation for this low incidence in comparison to the Neufeld at al. study is that we assessed patients in the PACU at 30 and 60 minutes following extubation of the endotracheal tube, and at the point of discharge from the PACU, so we were able to identify any possible changes in cognition and presence of delirium features. Neufeld et al. only conducted one assessment in the PACU once the patient reached a score of ≥ 9 on the Aldrete scoring system. Further, patients who were transferred to the ICU were included in their sample. This ICU population is known to have a high incidence of delirium, whereas in our study, we excluded patients who were transferred to the ICU. Nevertheless, the incidence of postoperative delirium in our study is consistent with the ranges described in the literature for postoperative delirium in patients admitted to the PACU.

Our finding regarding the diagnostic test accuracy of the 4AT is also supported by two previous studies. One study examined the 4AT in older patients admitted to the geriatric medical unit reported a sensitivity of 86.7% and reasonable specificity of 70%,1 while another study examined the diagnostic test accuracy of the 4AT in older patients admitted to an acute geriatric ward and rehabilitation department, and reported a sensitivity of 89.7% and specificity of 84%, which are similar to our reported values.26 Furthermore, a recent study by De et al.  emphasised the high values of diagnostic test accuracy of the 4AT, when conducted in older patients admitted to geriatric and orthogeriatric wards, where a sensitivity of 87% and specificity of 80% were reported.31